# Empirically Measuring Concentration: Fundamental Limits on Intrinsic Robustness

**Saeed Mahloujifar**[*]**, Xiao Zhang**[*]**, Mohammad Mahmoody, and David Evans**
University of Virginia
[saeed, shawn, mohammad, evans]@virginia.edu

## Abstract

Many recent works have shown that adversarial examples that fool classifiers can be found by minimally perturbing a normal input. Recent theoretical results, starting with Gilmer et al. (2018b), show that if the inputs are drawn from a *concentrated* metric probability space, then adversarial examples with small perturbation are inevitable. A concentrated space has the property that any subset with $\Omega(1)$ (e.g., 1/100) measure, according to the imposed distribution, has small distance to almost all (e.g., 99/100) of the points in the space. It is not clear, however, whether these theoretical results apply to actual distributions such as images. This paper presents a method for empirically measuring and bounding the concentration of a concrete dataset which is proven to converge to the actual concentration. We use it to empirically estimate the intrinsic robustness to $\ell_\infty$ and $\ell_2$ perturbations of several image classification benchmarks. Code for our experiments is available at https://github.com/xiaozhanguva/Measure-Concentration.

## 1   Introduction

Despite achieving exceptionally high accuracy on natural inputs, state-of-the-art machine learning models have been shown to be vulnerable to adversaries who use small perturbations to fool the classifier (Szegedy et al., 2014; Goodfellow et al., 2015). This phenomenon, known as *adversarial examples*, has motivated numerous studies (Papernot et al., 2016; Madry et al., 2018; Biggio & Roli, 2018; Gilmer et al., 2018a) to develop heuristic defenses that aim to improve classifier robustness. However, most defense mechanisms have been quickly broken by adaptive attacks (Carlini & Wagner, 2017; Athalye et al., 2018). Although certification methods (Raghunathan et al., 2018; Wong & Kolter, 2018; Sinha et al., 2018; Wong et al., 2018; Gowal et al., 2019; Wang et al., 2018; Zhang et al., 2019) have been proposed aiming to end such arms race and continuous efforts have been made to develop better robust models, both the robustness guarantees and efficiency achieved by state-of-the-art robust classifiers are far from satisfying.

This motivates a fundamental information-theoretic question: *what are the inherent limitations of developing robust classifiers?* Several recent works (Gilmer et al., 2018b; Fawzi et al., 2018; Mahloujifar et al., 2019; Shafahi et al., 2019; Bhagoji et al., 2019) have shown that under certain assumptions regarding the data distribution and the perturbation metric, adversarial examples are theoretically inevitable. As a result, for a broad set of theoretically natural metric probability spaces of inputs, there is no classifier for the data distribution that achieves adversarial robustness. For example, Gilmer et al. (2018b) assumed that the input data are sampled uniformly from $n$-spheres and proved a model-independent theoretical bound connecting the risk to the average Euclidean distance to the "caps" (i.e., round regions on a sphere). Mahloujifar et al. (2019) generalized this result to any concentrated metric probability space of inputs and showed, for example, that if the inputs come from

---

[*]Equal contribution.

any Normal Lévy family (Lévy, 1951), any classifier with a noticeable test error will be vulnerable to small (i.e., sublinear in the typical norm of the inputs) perturbations.

Although such theoretical findings seem discouraging to the goal of developing robust classifiers, all these impossibility results depend on assumptions about data distributions that might not hold for cases of interest. Our work develops a general method for testing properties of concrete datasets against these theoretical assumptions.

**Contributions.** Our work shrinks the gap between theoretical analyses of robustness of classification for theoretical data distributions and understanding the intrinsic robustness of actual datasets. Indeed, quantitative estimates of the intrinsic robustness[2] of benchmark image datasets such as MNIST and CIFAR-10 can provide us with a better understanding of the threat of adversarial examples for natural image distributions and may suggest promising directions for further improving classifier robustness. Our main technical contribution is a general method to evaluate the concentration of a given input distribution $\mu$ based on a set of data samples. We prove that by simultaneously increasing the sample size $m$ and a complexity parameter $T$, the concentration of the empirical measure converges to the actual concentration of $\mu$ (Section 3). Using this method, we perform experiments to demonstrate the existence of robust error regions for benchmark datasets under both $\ell_\infty$ and $\ell_2$ perturbations (Section 4). Compared with state-of-the-art robustly trained models, our estimated intrinsic robustness shows that, for most settings, there exists a large gap between the robust error achieved by the best current models and the theoretical limits implied by concentration. This suggests the concentration of measure is not the only reason behind the vulnerability of existing classifiers to adversarial perturbations. Thus, either there is room for improving the robustness of image classifiers (even with non-zero classification error) or a need for deeper understanding of the reasons for the gap between intrinsic robustness and the actual robustness achieved by robust models, at least for the datasets like the image classification benchmarks used in our experiments.

**Related Work.** We are aware of only one previous work that attempts to heuristically estimate these properties. To extend their theoretical impossibility result to the practical distributions, Gilmer et al. (2018b) studied MNIST dataset to find a region that is somewhat robust in terms of the *expected* $\ell_2$ distance of other images from the region. In their setting, they showed the existence of a set of measure $0.01$ with average $\ell_2$ distance $6.59$ to all points. In comparison, our work is the first to provide a general methodology to empirically estimate the concentration of measure with provable guarantees. Moreover, we are able to deal with $\ell_\infty$, and *worst-case* bounded perturbations for modeling adversarial risk, which is the most popular setting for research in adversarial examples. In addition, another related concurrent work (Bhagoji et al., 2019) studied lower bounds on the adversarial risk using optimal transport on the metric probability space of instances. They also measure the optimal transport on the empirical distributions but do not characterize the relationship between the optimal transport of empirical datasets and the actual one of the underlying distributions.

Another related line of work estimated lower bounds on the concentration of measure of the underlying distribution through simulating distributions by generative models. Fawzi et al. (2018) proved a lower bound on the concentration of the generated image distribution, assuming the underlying generative model has Gaussian latent space and small Lipschitz constant. Krusinga et al. (2019) estimated an upper bound on the density function of the distribution using generative model, then proved concentration inequalities based on upper bounds on the density function. Our work is distinct from these works, because we directly learn the concentration function instead of a lower bound, and we use the actual data samples instead of samples generated from some trained generative model.

The work of Tsipras et al. (2019) studied the trade-off between robustness and accuracy. They show that for some specific learning problems, achieving robustness and accuracy together is not possible. At first glance, it might seem that this trade-off contradicts the existing lower bounds that come from concentration of measure. However, there is no contradiction and what is proved there is with regard to a different definition of adversarial examples. The definition of adversarial examples used there could diverge from our definition in some learning problems (see Diochnos et al. (2018)), but they coincide in the cases that the ground truth function is robust to small perturbations.

**Notation.** Lowercase boldface letters such as $\boldsymbol{x}$ are used to denote vectors, and $[n]$ is used to represent $\{1, 2, \ldots, n\}$. For any set $\mathcal{A}$, let $\mathsf{Pow}(\mathcal{A})$, $|\mathcal{A}|$ and $\mathbb{1}_\mathcal{A}(\cdot)$ be the set of measurable subsets of $\mathcal{A}$, cardinality and indicator function of $\mathcal{A}$, respectively. For any $\boldsymbol{x} \in \mathbb{R}^n$, the $\ell_\infty$-norm and $\ell_2$-norm of $\boldsymbol{x}$ are defined as $\|\boldsymbol{x}\|_\infty = \max_{i \in [n]} |x_i|$ and $\|\boldsymbol{x}\|_2 = (\sum_{i \in [n]} x_i^2)^{1/2}$ respectively. Let $(\mathcal{X}, \mu)$ be a probability space and $d: \mathcal{X} \times \mathcal{X} \to \mathbb{R}$ be some distance metric defined on $\mathcal{X}$. Define the empirical measure with respect to a set $\mathcal{S}$ sampled from $\mu$ as $\hat{\mu}_\mathcal{S}(\mathcal{A}) = \sum_{\boldsymbol{x} \in \mathcal{S}} \mathbb{1}_\mathcal{A}(\boldsymbol{x})/|\mathcal{S}|, \forall \mathcal{A} \subseteq \mathcal{X}$. Let $\mathrm{Ball}(\boldsymbol{x}, \epsilon) = \{\boldsymbol{x}' \in \mathcal{X} : d(\boldsymbol{x}', \boldsymbol{x}) \leq \epsilon\}$ be the ball around $\boldsymbol{x}$ with radius $\epsilon$. For any subset $\mathcal{A} \subseteq \mathcal{X}$, define the $\epsilon$-expansion $\mathcal{A}_\epsilon = \{\boldsymbol{x} \in \mathcal{X} : \exists\, \boldsymbol{x}' \in \mathrm{Ball}(\boldsymbol{x}, \epsilon) \cap \mathcal{A}\}$. The collection of the $\epsilon$-expansions for members of any $\mathcal{G} \subseteq \mathsf{Pow}(\mathcal{X})$ is defined and denoted as $\mathcal{G}_\epsilon = \{\mathcal{A}_\epsilon : \mathcal{A} \in \mathcal{G}\}$.

## 2 Robustness and Concentration of Measure

In this paper, we work with the following definition of *adversarial risk*:

**Definition 2.1** (Adversarial Risk). *Let $(\mathcal{X}, \mu)$ be the probability space of instances and $f^*$ be the underlying ground-truth. The* adversarial risk *of a classifier $f$ in metric $d$ with strength $\epsilon$ is defined as*

$$\mathrm{AdvRisk}_\epsilon(f, f^*) = \Pr_{\boldsymbol{x} \leftarrow \mu}\left[\exists\, \boldsymbol{x}' \in \mathrm{Ball}(\boldsymbol{x}, \epsilon) \text{ s.t. } f(\boldsymbol{x}') \neq f^*(\boldsymbol{x}')\right].^3$$

For $\epsilon = 0$, which allows no perturbation, the notion of adversarial risk coincides with traditional risk.

**Definition 2.2** (Intrinsic Robustness). *Consider the same setting as in Definition 2.1. Let $\mathcal{F}$ be some family of classifiers, then the* intrinsic robustness *is defined as the maximum adversarial robustness that can be achieved within $\mathcal{F}$, namely*

$$\mathrm{Rob}_\epsilon(\mathcal{F}, f^*) = 1 - \inf_{f \in \mathcal{F}}\left\{\mathrm{AdvRisk}_\epsilon(f, f^*)\right\}.$$

In this work, we specify $\mathcal{F}$ as the family of imperfect classifiers that have risk at least $\alpha \in (0, 1)$.

Previous work shows a connection between concentration of measure and the intrinsic robustness with respect to some families of classifiers (Gilmer et al. (2018b); Fawzi et al. (2018); Mahloujifar et al. (2019); Shafahi et al. (2019)). The concentration of measure on a metric probability space is defined by a concentration function as follows.

**Definition 2.3** (Concentration Function). *Consider a metric probability space $(\mathcal{X}, \mu, d)$. Suppose $\epsilon > 0$ and $\alpha \in (0, 1)$ are given parameters, then the* concentration function *of the probability measure $\mu$ with respect to $\epsilon$, $\alpha$ is defined as*

$$h(\mu, \alpha, \epsilon) = \inf_{\mathcal{E} \in \mathsf{Pow}(\mathcal{X})}\left\{\mu(\mathcal{E}_\epsilon) : \mu(\mathcal{E}) \geq \alpha\right\}.$$

Note that the standard notion of concentration function (e.g., see Talagrand (1995)) is related to a special case of Definition 2.3 by fixing $\alpha = 1/2$.

Generalizing the result of Gilmer et al. (2018b) about instances drawn from spheres, Mahloujifar et al. (2019) showed that, in general, if the metric probability space of instances is concentrated, then any classifier with 1% risk incurs large adversarial risk for small amount of perturbations.

**Theorem 2.4** (Mahloujifar et al. (2019)). *Let $(\mathcal{X}, \mu)$ be the probability space of instances and $f^*$ be the underlying ground-truth. For any classifier $f$, we have*

$$\mathrm{AdvRisk}_\epsilon(f, f^*) \geq h(\mu, \mathrm{Risk}(f, f^*), \epsilon).$$

In order for this theorem to be useful, we need to know the concentration function. The behavior of this function is studied extensively for certain theoretical metric probability spaces (Ledoux, 2001; Milman & Schechtman, 1986). However, it is not known how to measure the concentration function for arbitrary metric probability spaces. In this work, we provide a framework to (algorithmically) bound the concentration function from i.i.d. samples from a distribution. Namely, we want to solve the following optimization task using our i.i.d. samples:

$$\underset{\mathcal{E} \in \mathsf{Pow}(\mathcal{X})}{\text{minimize}}\ \mu(\mathcal{E}_\epsilon) \quad \text{subject to } \mu(\mathcal{E}) \geq \alpha. \tag{1}$$

We aim to estimate the minimum possible adversarial risk, which captures the intrinsic robustness for classification in terms of the underlying distribution $\mu$, conditioned on the fact that the original risk is at least $\alpha$. Note that solving this optimization problem only shows the possibility of existence of an error region $\mathcal{E}$ with certain (small) expansion. This means that there could potentially exist a classifier with risk at least $\alpha$ and adversarial risk equal to the solution of the optimization problem of (1). Actually *finding* such an optimally robust classifier (with error $\alpha$) using a learning algorithm might be a much more difficult task or even infeasible. We do not consider that problem in this work.

## 3  Method for Measuring Concentration

In this section, we present a method to measure the concentration of measure on a metric probability space using i.i.d. samples. To measure concentration, there are two main challenges:

1. Measuring concentration appears to require knowledge of the density function of the distribution, but we only have a data set sampled from the distribution.

2. Even with the density function, we have to find the best possible subset among all the subsets of the space, which seems infeasible.

We show how to overcome these challenges and find the actual concentration in the limit by first empirically simulating the distribution and then narrowing down our search space to a specific collection of subsets. Our results show that for a carefully chosen family of sets, the set with minimum expansion can be approximated using polynomially many samples. On the other hand, the minimum expansion convergence to the actual concentration (without the limits on the sets) as the complexity of the collection goes to infinity.

Before stating our main theorems, we introduce two useful definitions. The following definition captures the concentration function for a specific collection of subsets.

**Definition 3.1** (Concentration Function for a Collection of Subsets). *Consider a metric probability space $(\mathcal{X}, \mu, d)$. Let $\epsilon > 0$ and $\alpha \in (0, 1)$ be given parameters, then the* concentration function *of the probability measure $\mu$ with respect to $\epsilon$, $\alpha$ and a collection of subsets $\mathcal{G} \subseteq \mathsf{Pow}(\mathcal{X})$ is defined as*

$$h(\mu, \alpha, \epsilon, \mathcal{G}) = \inf_{\mathcal{E} \in \mathcal{G}} \{\mu(\mathcal{E}_\epsilon) \colon \mu(\mathcal{E}) \geq \alpha\}.$$

*When $\mathcal{G} = \mathsf{Pow}(\mathcal{X})$, we write $h(\mu, \alpha, \epsilon)$ for simplicity.*

We also need to define the notion of complexity penalty for a collection of subsets. The complexity penalty for a collection of subsets captures the rate of the uniform convergence for the subsets in that collection. One can get such uniform convergence rates using the VC dimension or Rademacher complexity of the collection.

**Definition 3.2** (Complexity Penalty). *Let $\mathcal{G} \subseteq \mathsf{Pow}(\mathcal{X})$ be a collection of subsets of $\mathcal{X}$. A function $\phi \colon \mathbb{N} \times \mathbb{R} \to [0, 1]$ is a complexity penalty for $\mathcal{G}$ iff for any probability measure $\mu$ supported on $\mathcal{X}$ and any $\delta \in [0, 1]$, we have*

$$\Pr_{S \leftarrow \mu^m}[\exists\, \mathcal{E} \in \mathcal{G}\ \ s.t.\ \ |\mu(\mathcal{E}) - \hat{\mu}_S(\mathcal{E})| \geq \delta] \leq \phi(m, \delta).$$

Theorem 3.3 shows how to overcome the challenge of measuring concentration from finite samples, when the concentration is defined with respect to specific families of subsets. Namely, it shows that the empirical concentration is close to the true concentration, if the underlying collection of subsets is not too complex. The proof of Theorem 3.3 is provided in Appendix A.1.

**Theorem 3.3** (Generalization of Concentration). *Let $(\mathcal{X}, \mu, d)$ be a metric probability space and $\mathcal{G} \subseteq \mathsf{Pow}(\mathcal{X})$. For any $\delta, \alpha, \epsilon \in [0, 1]$, we have*

$$\Pr_{S \leftarrow \mu^m}[h(\mu, \alpha - \delta, \epsilon, \mathcal{G}) - \delta \leq h(\hat{\mu}_S, \alpha, \epsilon, \mathcal{G}) \leq h(\mu, \alpha + \delta, \epsilon, \mathcal{G}) + \delta] \geq 1 - 2\big(\phi(m, \delta) + \phi_\epsilon(m, \delta)\big)$$

*where $\phi$ and $\phi_\epsilon$ are complexity penalties for $\mathcal{G}$ and $\mathcal{G}_\epsilon$ respectively.*

**Remark 3.4.** Theorem 3.3 shows that if we narrow down our search to a collection of subsets $\mathcal{G}$ such that both $\mathcal{G}$ and $\mathcal{G}_\epsilon$ have small complexity penalty, then we can use the empirical distribution to measure concentration of measure for that specific collection. Note that the generalization bound of

Theorem 3.3 depends on complexity penalties for both $\mathcal{G}$ and $\mathcal{G}_\epsilon$. Therefore, in order for this theorem to be useful, the collection $\mathcal{G}$ must be chosen in a careful way. For example, if $\mathcal{G}$ has bounded VC dimension, then $\mathcal{G}_\epsilon$ might still have a very large VC dimension. Alternatively, $\mathcal{G}$ might denote the collection of subsets that are decidable by a neural network of a certain size. In that case, even though there are well known complexity penalties for such collections (see Neyshabur et al. (2017)), the complexity of their *expansions* is unknown. In fact, relating the complexity penalty for expansion of a collection to that of the original collection is tightly related to generalization bounds in the adversarial settings, which has also been the subject of several recent works (Cullina et al., 2018; Attias et al., 2019; Montasser et al., 2019; Yin et al., 2019; Raghunathan et al., 2019).

The following theorem, proved in Appendix A.2, states that if we gradually increase the complexity of the collection and the number of samples together, the empirical estimate of concentration converges to actual concentration, as long as several conditions hold. Theorem 3.5 and the techniques used in its proof are inspired by the work of Scott & Nowak (2006) on learning minimum volume sets.

**Theorem 3.5.** *Let* $\{\mathcal{G}(T)\}_{T\in\mathbb{N}}$ *be a family of subset collections defined over a space* $\mathcal{X}$*. Let* $\{\phi^T\}_{T\in\mathbb{N}}$ *and* $\{\phi_\epsilon^T\}_{T\in\mathbb{N}}$ *be two families of complexity penalty functions such that* $\phi^T$ *and* $\phi_\epsilon^T$ *are complexity penalties for* $\mathcal{G}(T)$ *and* $\mathcal{G}_\epsilon(T)$ *respectively, for some* $\epsilon \in [0,1]$*. Let* $\{m(T)\}_{T\in\mathbb{N}}$ *and* $\{\delta(T)\}_{T\in\mathbb{N}}$ *be two sequences such that* $m(T) \in \mathbb{N}$ *and* $\delta(T) \in [0,1]$*.*

*Consider a sequence of datasets* $\{S_T\}_{T\in\mathbb{N}}$*, where* $S_T$ *consists of* $m(T)$ *i.i.d. samples from a measure* $\mu$ *supported on* $\mathcal{X}$*. Also let* $\alpha \in [0,1]$ *be such that* $h$ *is locally continuous w.r.t the second parameter at point* $(\mu, \alpha, \epsilon, \mathsf{Pow}(\mathcal{X}))$*. If all the following hold,*

1. $\sum_{T=1}^\infty \phi^T(m(T), \delta(T)) < \infty$

2. $\sum_{T=1}^\infty \phi_\epsilon^T(m(T), \delta(T)) < \infty$

3. $\lim_{T\to\infty} \delta(T) = 0$

4. $\lim_{T\to\infty} h(\mu, \alpha, \epsilon, \mathcal{G}(T)) = h(\mu, \alpha, \epsilon)$

*then with probability 1, we have* $\lim_{T\to\infty} h(\hat{\mu}_{S_T}, \alpha, \epsilon, \mathcal{G}(T)) = h(\mu, \alpha, \epsilon)$*.*

**Remark 3.6.** In Theorem 3.5, the first two conditions restrict the growth rate for the complexity of the collections. Namely, we need the complexity penalties $\phi^T(m(T), \delta(T))$ and $\phi_\epsilon^T(m(T), \delta(T))$ to rapidly approach 0 as $T \to \infty$, which means the complexity of $\mathcal{G}(T)$ and $\mathcal{G}_\epsilon(T)$ should grow at a slow rate. The third condition requires that our generalization error goes to zero as we increase $T$. Note that the complexity penalty is a decreasing function with respect to $\delta$, which means condition 3 makes achieving the first two conditions harder. However, since the complexity penalty is a function of both $\delta$ and sample size, we can still increase the sample size with a faster rate to satisfy the first two conditions. Finally, the fourth condition requires our approximation error goes to 0 as we increase $T$. Note that this condition holds for any family of collections of subsets that is a universal approximator (e.g., decision trees or neural networks). However, in order for our theorem to hold, we also need all the other conditions. In particular, we cannot use decision trees or neural networks as our collection of subsets, because we do not know if there is a complexity penalty for them that satisfies condition 2.

## 3.1 Special Case of $\ell_\infty$

In this subsection, we show how to instantiate Theorem 3.5 for the case of $\ell_\infty$. Below, we introduce a special collection of subsets characterized by the *complement of a union of hyperrectangles*:

**Definition 3.7** (Complement of union of hyperrectangles)**.** *For any positive integer* $T$*, the collection of subsets specified by the* complement of a union of $T$ $n$-dimensional hyperrectangles *is defined as*

$$\mathcal{CR}(T,n) = \left\{ \mathbb{R}^n \setminus \cup_{t=1}^T \mathcal{R}ect(\boldsymbol{u}^{(t)}, \boldsymbol{r}^{(t)}) \colon \forall t \in [T], (\boldsymbol{u}^{(t)}, \boldsymbol{r}^{(t)}) \in \mathbb{R}^n \times \mathbb{R}_{\geq 0}^n \right\},$$

*where* $\mathcal{R}ect(\boldsymbol{u}, \boldsymbol{r}) = \{\boldsymbol{x} \in \mathcal{X} : \forall j \in [n], |x_j - u_j| \leq r_j/2\}$ *denotes the hyperrectangle centered at* $\boldsymbol{u}$ *with* $\boldsymbol{r}$ *representing the edge size vector. When* $n$ *is free of context, we simply write* $\mathcal{CR}(T)$*.*

Recall that our goal is to find a subset $\mathcal{E} \in \mathbb{R}^n$ such that $\mathcal{E}$ has measure at least $\alpha$ and the $\epsilon_\infty$-expansion of $\mathcal{E}$ under $\ell_\infty$ has the minimum measure. To achieve this goal, we approximate the distribution $\mu$

with an empirical distribution $\hat{\mu}_S$, and limit our search to the special collection $\mathcal{CR}(T)$ (though our goal is to find the minimum concentration around arbitrary subsets). Namely, what we find is still an *upper bound* on the concentration function, and it is an upper bound that we know it converges the actual value in the limit. Our problem thus becomes the following optimization task:

$$\underset{\mathcal{E} \in \mathcal{CR}(T)}{\text{minimize}} \ \hat{\mu}_S(\mathcal{E}_{\epsilon_\infty}) \quad \text{subject to } \hat{\mu}_S(\mathcal{E}) \geq \alpha. \tag{2}$$

The following theorem provides the key to our empirical method by providing a convergence guarantee. It states that if we increase the number of rectangles and the number of samples together in a careful way, the solution to the problem using restricted sets converges to the true concentration.

**Theorem 3.8.** *Consider a nice metric probability space* $(\mathbb{R}^n, \mu, \ell_\infty)$. *Let* $\{S_T\}_{T \in \mathbb{N}}$ *be a family of datasets such that for all* $T \in \mathbb{N}$, $S_T$ *contains at least* $T^4$ *i.i.d. samples from* $\mu$. *For any* $\epsilon_\infty$ *and* $\alpha \in [0, 1]$, *if* $h$ *is locally continuous w.r.t the second parameter at point* $(\mu, \alpha, \epsilon_\infty)$, *then with probability* 1 *we get*

$$\lim_{T \to \infty} h(\hat{\mu}_{S_T}, \alpha, \epsilon_\infty, \mathcal{CR}(T)) = h(\mu, \alpha, \epsilon_\infty).$$

Note that the size of $S_T$ is selected as $T^4$ to guarantee conditions 1 and 2 are satisfied in Theorem 3.5. In fact, we can tune the parameters more carefully to get $T^2$, instead of $T^4$, but the convergence will be slower. See Appendix A.3 for the proof.

## 3.2 Special Case of $\ell_2$

This subsection demonstrates how to apply Theorem 3.5 to the case of $\ell_2$. The following definition introduces the collection of subsets characterized by a *union of balls*:

**Definition 3.9** (Union of Balls). *For any positive integer* $T$, *the collection of subsets specified by a* union of $T$ $n$-dimensional balls *is defined as*

$$\mathcal{B}(T, n) = \left\{ \cup_{t=1}^T \text{Ball}(\boldsymbol{u}^{(t)}, \boldsymbol{r}^{(t)}) \colon \forall t \in [T], (\boldsymbol{u}^{(t)}, \boldsymbol{r}^{(t)}) \in \mathbb{R}^n \times \mathbb{R}_{\geq 0}^n \right\}.$$

*When* $n$ *is free of context, we simply write* $\mathcal{B}(T)$.

By restricting our search to the collection of a union of balls $\mathcal{B}(T)$ and replacing the underlying distribution $\mu$ with the empirical one $\hat{\mu}_S$, our problem becomes the following optimization task

$$\underset{\mathcal{E} \in \mathcal{B}(T)}{\text{minimize}} \ \hat{\mu}_S(\mathcal{E}_{\epsilon_2}) \quad \text{subject to } \hat{\mu}_S(\mathcal{E}) \geq \alpha. \tag{3}$$

Theorem 3.10, proven in Appendix A.4, guarantees that if we increase the number of balls and samples together in a careful way, the solution to the empirical problem (3) converges to the true concentration.

**Theorem 3.10.** *Consider a nice metric probability space* $(\mathbb{R}^n, \mu, \ell_2)$. *Let* $\{S_T\}_{T \in \mathbb{N}}$ *be a family of datasets such that for all* $T \in \mathbb{N}$, $S_T$ *contains at least* $T^4$ *i.i.d. samples from* $\mu$. *For any* $\epsilon_2$ *and* $\alpha \in [0, 1]$, *if* $h$ *is locally continuous w.r.t the second parameter at point* $(\mu, \alpha, \epsilon_2)$, *then with probability* 1 *we get*

$$\lim_{T \to \infty} h(\hat{\mu}_{S_T}, \alpha, \epsilon_2, \mathcal{B}(T)) = h(\mu, \alpha, \epsilon_2).$$

# 4 Experiments

In this section, we provide heuristic methods to find the best possible error region, which covers at least $\alpha$ fraction of the samples and its expansion covers the least number of points, for both $\ell_\infty$ and $\ell_2$ settings. Specifically, we first introduce our algorithm, then evaluate our approach on two benchmark image datasets: MNIST (LeCun et al., 2010) and CIFAR-10 (Krizhevsky & Hinton, 2009). Note that in our experiments we exactly use the collection of subsets as suggested by our theoretical results in the previous section. However, that is not necessary and one might work with any subset collection to run experiments, as long as they can estimate the measure of the sets and their expansion. We tried working with other collection of subsets that we do not have theoretical support for (e.g. sets defined by a neural network) and observed a large generalization gap. This observation shows the importance of working with subset collections that we can theoretically control their generalization penalty.

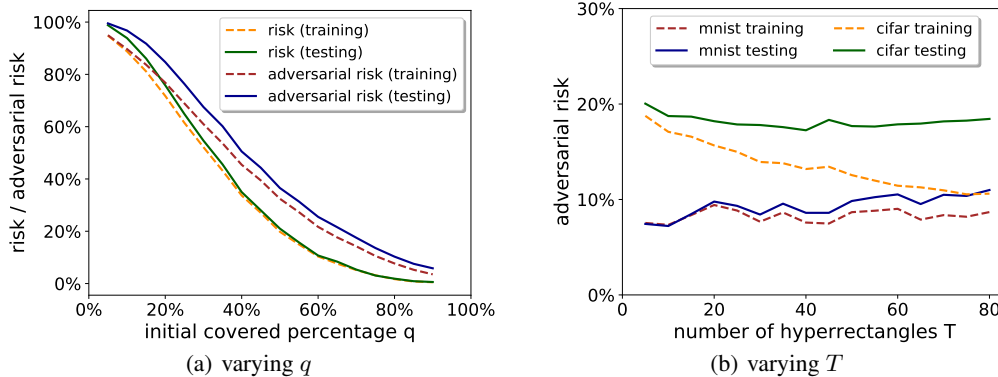

(a) varying $q$                    (b) varying $T$

Figure 1: (a) Plots of risk and adversarial risk w.r.t. the resulted error region using our method as $q$ varies (CIFAR-10, $\epsilon_\infty = 8/255$, $T = 30$); (b) Plots of adversarial risk w.r.t. the resulted error region using our method (best $q$) as $T$ varies on MNIST ($\epsilon_\infty = 0.3$) and CIFAR-10 ($\epsilon_\infty = 8/255$).

## 4.1 Experiments for $\ell_\infty$

Theorem 3.8 shows that the empirical concentration function $h(\hat{\mu}_S, \alpha, \epsilon_\infty, \mathcal{CR}(T))$ converges to the actual concentration $h(\mu, \alpha, \epsilon_\infty)$ asymptotically, when $T$ and $|\mathcal{S}|$ go to infinity with $|\mathcal{S}| \geq T^4$. Thus, to measure the concentration of $\mu$, it remains to solve the optimization problem (2).

**Method.** Although the collection of subsets is specified using simple topology, solving (2) exactly is still difficult, as the problem itself is combinatorial in nature. Borrowing techniques from clustering, we propose an empirical method to search for desirable error region within $\mathcal{CR}(T)$. Any error region $\mathcal{E}$ could be used to define $f_\mathcal{E}$, i.e., $f_\mathcal{E}(\boldsymbol{x}) = f^*(\boldsymbol{x})$, if $\boldsymbol{x} \notin \mathcal{E}$; $f_\mathcal{E}(\boldsymbol{x}) \neq f^*(\boldsymbol{x})$, if $\boldsymbol{x} \in \mathcal{E}$. However, finding a classifier corresponding to $f_\mathcal{E}$ using a learning algorithm might be a very difficult task. Here, we find the optimally robust error region, not the corresponding classifier. A desirable error region should have small adversarial risk[4], compared with all subsets in $\mathcal{CR}(T)$ that have measure at least $\alpha$.

The high-level intuition is that images from different classes are likely to be concentrated in separable regions, since it is generally believed that small perturbations preserve the ground-truth class at the sampled images. Therefore, if we cluster all the images into different clusters, a desired region with low adversarial risk should exclude any image from the dense clusters, otherwise the expansion of such a region will quickly cover the whole cluster. In other words, a desirable subset within $\mathcal{CR}(T)$ should be $\epsilon_\infty$ away (in $\ell_\infty$ norm) from all the dense image clusters, which motivates our method to cover the dense image clusters using hyperrectangles and treat the complement of them as error set.

More specifically, our algorithm (for pseudocode, see Algorithm 1 in Appendix B) starts by sorting all the training images in an ascending order based on the $\ell_1$-norm distance to the $k$-th nearest neighbour with $k = 50$, and then obtains $T$ hyperrectangular image clusters by performing $k$-means clustering (Hartigan & Wong, 1979) on the top-$q$ densest images, where the metric is chosen as $\ell_1$ and the maximum iterations is set as 30. Finally, we perform a binary search over $q \in [0, 1]$, where we set $\delta_{\text{bin}} = 0.005$ as the stopping criteria, to obtain the best robust subset (lowest adversarial risk) in $\mathcal{CR}(T)$ with empirical measure at least $\alpha$.

**Results.** We choose $\alpha$ to reflect the best accuracy achieved by state-of-the-art classifiers, using $\alpha = 0.01$ and $\epsilon_\infty \in \{0.1, 0.2, 0.3, 0.4\}$ for MNIST and selecting appropriate values to represent the best typical results on the other datasets (see Table 1). Given the number of hyperrectangles, $T$, we obtain the resulting error region using the proposed algorithm on the training dataset, and tune $T$ for the minimum adversarial risk on the testing dataset.

Figure 1 shows the learning curves regarding risk and adversarial risk for two specific experimental settings (similar results are obtained under other experimental settings, see Appendix C.3). Figure 1(a) suggests that as we increase the initial covered percentage $q$, both risk and adversarial risk of the corresponding error region decrease. This supports our use of binary search on $q$ in Algorithm 1. On

Table 1: Summary of the main results using our method for different settings with $\ell_\infty$ perturbations.

| Dataset | $\alpha$ | $\epsilon_\infty$ | $T$ | Best $q$ | Empirical Risk (%) | | Empirical AdvRisk (%) | |
|---|---|---|---|---|---|---|---|---|
| | | | | | training | testing | training | testing |
| MNIST | 0.01 | 0.1 | 5 | 0.662 | $1.22 \pm 0.11$ | $1.23 \pm 0.12$ | $3.65 \pm 0.29$ | $3.64 \pm 0.30$ |
| | | 0.2 | 10 | 0.660 | $1.12 \pm 0.13$ | $1.11 \pm 0.10$ | $5.76 \pm 0.38$ | $5.89 \pm 0.44$ |
| | | 0.3 | 10 | 0.629 | $1.12 \pm 0.12$ | $1.15 \pm 0.13$ | $7.34 \pm 0.38$ | $7.24 \pm 0.38$ |
| | | 0.4 | 10 | 0.598 | $1.15 \pm 0.09$ | $1.21 \pm 0.09$ | $9.89 \pm 0.57$ | $9.92 \pm 0.60$ |
| CIFAR-10 | 0.05 | 2/255 | 10 | 0.680 | $5.32 \pm 0.21$ | $5.72 \pm 0.25$ | $7.29 \pm 0.20$ | $8.13 \pm 0.26$ |
| | | 4/255 | 20 | 0.688 | $5.59 \pm 0.25$ | $6.05 \pm 0.40$ | $11.43 \pm 0.24$ | $13.66 \pm 0.33$ |
| | | 8/255 | 40 | 0.734 | $5.55 \pm 0.21$ | $5.94 \pm 0.34$ | $13.69 \pm 0.19$ | $18.13 \pm 0.30$ |
| | | 16/255 | 75 | 0.719 | $5.16 \pm 0.25$ | $5.28 \pm 0.23$ | $19.77 \pm 0.22$ | $28.83 \pm 0.46$ |

Table 2: Comparisons between our method and the existing adversarially trained robust classifiers under different settings. We use the *Risk* and *AdvRisk* for robust training methods to denote the standard test error and attack success rate reported in literature. The *AdvRisk* reported for our method can be seen as an estimated lower bound of adversarial risk for existing classifiers.

| Dataset | Strength (metric) | Method | Empirical Risk | Empirical AdvRisk |
|---|---|---|---|---|
| MNIST | $\epsilon_\infty = 0.3$ | Madry et al. (2018) | 1.20% | 10.70% |
| | | Ours ($T = 10, \alpha = 0.012$) | $1.35\% \pm 0.08\%$ | $8.28\% \pm 0.22\%$ |
| MNIST | $\epsilon_2 = 1.5$ | Schott et al. (2019) | 1.00% | 20.00% |
| | | Ours ($T = 20, \alpha = 0.01$) | 1.08% | 2.12% |
| CIFAR-10 | $\epsilon_\infty = 8/255$ | Madry et al. (2018) | 12.70% | 52.96% |
| | | Ours ($T = 40, \alpha = 0.127$) | $14.22\% \pm 0.46\%$ | $29.21\% \pm 0.35\%$ |

the other hand, as can be seen from Figure 1(b), overfitting with respect to adversarial risk becomes significant as we increase the number of hyperrectangles. According to the adversarial risk curve for testing data, the optimal value of $T$ is selected as $T = 10$ for MNIST ($\epsilon_\infty = 0.3$) and $T = 40$ for CIFAR-10 ($\epsilon_\infty = 8/255$).

Table 1 summarizes the optimal parameters, the empirical risk and adversarial risk of the corresponding error region on both training and testing datasets for each experimental setting (see Appendix C.1 for similar results on Fashion-MNIST and SVHN). Since the $k$-means algorithm does not guarantee global optimum, we repeat our method for 10 runs with random restarts in terms of the best parameters, then report both the mean and the standard deviation. Our experiments provide examples of rather robust error regions for real image datasets. For instance, in Table 1 we have a case where the measure of the resulting error region increases from 5.94% to 18.13% after expansion with $\epsilon_\infty = 8/255$ on CIFAR-10 dataset. This means that there could potentially be a classifier with 5.94% risk and 18.13% adversarial risk, but the-state-of-the-art robust classifier has empirically-measured adversarial risk 52.96% (Madry et al., 2018).

Noticing that the risk lower threshold $\alpha = 0.05$ is much lower than the empirical risk 12.70% of the adversarially-trained robust model reported in Madry et al. (2018), we further measure the empirical concentration on MNIST and CIFAR-10 using our method with $\alpha$ set to be the same as the reported standard test error in Madry et al. (2018), which is demonstrated in Table 2. In particular, we show that the gap between the attack success rate of Madry et al.'s classifier (10.70%) and our estimated best-achievable adversarial risk (8.28%) is quite small on MNIST, suggesting that the robustness of Madry et al.'s classifier is actually close to the intrinsic robustness. In sharp contrast, the gap becomes significantly larger on CIFAR-10: 29.21% for our estimate, while 52.96% for the reported attack success rate in Madry et al. (2018). Regardless of the difference, this gap cannot be explained by the concentration of measure phenomenon, suggesting there may still be room for developing more robust classifiers, or that other inherent reasons impede learning a more robust classifier.

Table 3: Comparisons between different methods for finding robust error region with $\ell_2$ perturbations.

| Dataset | $\alpha$ | $\epsilon_2$ | Gilmer et al. (2018b) | | Our Method | | |
|---------|----------|--------------|------|---------|---|------|---------|
| | | | Risk | AdvRisk | $T$ | Risk | AdvRisk |
| MNIST | 0.01 | 1.58 | 1.18% | 3.92% | 20 | 1.07% | 2.19% |
| | | 3.16 | 1.18% | 9.73% | 20 | 1.02% | 4.15% |
| | | 4.74 | 1.18% | 23.40% | 20 | 1.07% | 10.09% |
| CIFAR-10 | 0.05 | 0.2453 | 5.27% | 5.58% | 5 | 5.16% | 5.53% |
| | | 0.4905 | 5.27% | 5.93% | 5 | 5.14% | 5.83% |
| | | 0.9810 | 5.27% | 6.47% | 5 | 5.12% | 6.56% |

## 4.2 Experiments for $\ell_2$

For $\ell_2$ adversaries, Theorem 3.10 guarantees the asymptotic convergence of the empirical concentration function characterized by union of balls $\mathcal{B}(T)$ towards the actual concentration. Thus, it remains to solve the corresponding optimization problem (3). Similar to $\ell_\infty$, we propose an empirical method to search for desirable robust error regions under $\ell_2$ perturbations. From a high level, our algorithm (for pseudocode, see Algorithm 2 in Appendix B) places $T$ balls in a sequential manner, and searches for the best possible placement using a greedy approach at each time. Since enumerating all the possible ball centers is infeasible, we restrict the choice of the center to be the set of training data points. Our method keeps two sets of indices: one for the initial coverage and one for the coverage after expansion, and updates them when we find the optimal placement, i.e. the ball centered at some training data point that has the minimum expansion with respect to both sets.

We compare our empirical method for finding robust error regions characterized by a union of balls with the hyperplane-based approach (Gilmer et al., 2018b) on MNIST and CIFAR-10. In particular, the risk threshold $\alpha$ is set to be the same as the case of $\ell_\infty$, and the adversarial strength $\epsilon_2$ is chosen such that the volume of an $\ell_2$ ball with radius $\epsilon_2$ is roughly the same as the $\ell_\infty$ ball with radius $\epsilon_\infty$, using the conversion rule $\epsilon_2 = \sqrt{n/\pi} \cdot \epsilon_\infty$ as in Wong et al. (2018). Table 3 summarizes the optimal parameters, the testing risk and adversarial risk (see Appendix C.2 for more detailed results, including for other datasets) of the trained error regions using different methods, where we tune the number of balls $T$ for our method.

Our results show that there exist rather robust $\ell_2$ error regions for real image datasets. For example, the measure of the resulting error region using our method only increases by 0.69% (from 5.14% to 5.83%) after expansion with $\epsilon_2 = 0.4905$ on CIFAR-10. Compared with Gilmer et al. (2018b), our method is able to find regions with significantly smaller adversarial risk (around half the adversarial risk of regions found by their method) on MNIST, while attaining comparable error region robustness on CIFAR-10. Nevertheless, the adversarial risk attained by state-of-the-art robust classifiers against $\ell_2$ perturbations is much higher than these reported rates (see Table 2 for a comparison with the best robust classifier against $\ell_2$ perturbations proposed in Schott et al. (2019)).

## 5 Conclusion

To understand whether theoretical results showing limits of intrinsic robustness for natural distributions apply to concrete datasets, we developed a general framework to measure the concentration of an unknown distribution through its i.i.d. samples and a carefully-selected collection of subsets. Our experimental results suggest that the concentration of measure phenomenon is not the sole reason behind vulnerability of the existing classifiers to adversarial examples. In other words, recent impossibility results (Gilmer et al., 2018b; Fawzi et al., 2018; Mahloujifar et al., 2019; Shafahi et al., 2019) should not cause us to lose hope in the possibility of finding more robust classifiers.

**Acknowledgements.** This work was partially funded by an award from the National Science Foundation SaTC program (Center for Trustworth Machine Learning, #1804603), an NSF CAREER award (CCF-1350939), and support from Baidu, Intel, and Amazon.

## Footnotes

[2]See Definition 2.2 for the formal definition of intrinsic robustness. The term robustness has been used with different meanings in previous works (e.g., in Diochnos et al. (2018), it refers to the average distances to the error region). However, all such uses refer to a desirable property of the classifier in being resilient to adversarial perturbations, which is the case here as well. See Diochnos et al. (2018) for a taxonomy of different definitions.

[3]Note that bounding $l_p$ norm might be restrictive for the adversary (Gilmer et al., 2018a) and this definition only covers a subset of possible adversaries.

[4]The adversarial risk of an error region $\mathcal{E}$ simply refers to the adversarial risk of $f_\mathcal{E}$.

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
