[Supplementary Material]

# A Proofs of Theorems in Section 3

In this section, we prove Theorems 3.3, 3.5, 3.8 and 3.10.

## A.1 Proof of Theorem 3.3

*Proof.* Define $g(\mu, \alpha, \epsilon, \mathcal{G}) = \operatorname{argmin}_{\mathcal{E} \in \mathcal{G}} \{\mu(\mathcal{E}_\epsilon) \colon \mu(\mathcal{E}) \geq \alpha\}$, and let $\mathcal{E} = g(\mu, \alpha + \delta, \epsilon, \mathcal{G})$ and $\hat{\mathcal{E}} = g(\hat{\mu}_S, \alpha, \epsilon, \mathcal{G})$. (Note that these sets achieving the minimum might not exist, in which case we select a set for which the expansion is arbitrarily close to the infimum and every step of the proof will extend to this variant).

By the definition of the complexity penalty we have

$$\Pr_{S \leftarrow \mu^m} \left[ \left| \mu(\hat{\mathcal{E}}) - \hat{\mu}_S(\hat{\mathcal{E}}) \right| \geq \delta \right] \leq \phi(m, \delta)$$

which implies

$$\Pr_{S \leftarrow \mu^m} [\mu(\hat{\mathcal{E}}) \leq \alpha - \delta] \leq \phi(m, \delta).$$

Therefore, by the definition of $h$ we have

$$\Pr_{S \leftarrow \mu^m} [\mu(\hat{\mathcal{E}}_\epsilon) \leq h(\mu, \alpha - \delta, \epsilon, \mathcal{G})] \leq \phi(m, \delta). \tag{4}$$

On the other hand, based on the definition of $\phi_\epsilon$ we have

$$\Pr_{S \leftarrow \mu^m} \left[ \left| \mu(\hat{\mathcal{E}}_\epsilon) - \hat{\mu}_S(\hat{\mathcal{E}}_\epsilon) \right| \geq \delta \right] \leq \phi_\epsilon(m, \delta). \tag{5}$$

Combining Equation 4 and Equation 5, and by a union bound we get

$$\Pr_{S \leftarrow \mu^m} [\hat{\mu}_S(\hat{\mathcal{E}}_\epsilon) \leq h(\mu, \alpha - \delta, \epsilon, \mathcal{G}) - \delta] \leq \phi(m, \delta) + \phi_\epsilon(m, \delta)$$

which by the definition of $\hat{\mathcal{E}}$ implies that

$$\Pr_{S \leftarrow \mu^m} [h(\hat{\mu}_S, \alpha, \epsilon, \mathcal{G}) \leq h(\mu, \alpha - \delta, \epsilon, \mathcal{G}) - \delta] \leq \phi(m, \delta) + \phi_\epsilon(m, \delta). \tag{6}$$

Now we bound the probability for the other side of our inequality. By the definition of the notion of complexity penalty we have

$$\Pr_{S \leftarrow \mu^m} [|\mu(\mathcal{E}) - \hat{\mu}_S(\mathcal{E})| \geq \delta] \leq \phi(m, \delta)$$

which implies

$$\Pr_{S \leftarrow \mu^m} [\hat{\mu}_S(\mathcal{E}) \leq \alpha] \leq \phi(m, \delta).$$

Therefore, by the definition of $h$ we have,

$$\Pr_{S \leftarrow \mu^m} [\hat{\mu}_S(\mathcal{E}_\epsilon) \leq h(\hat{\mu}_S, \alpha, \epsilon, \mathcal{G})] \leq \phi(m, \delta). \tag{7}$$

On the other hand, based on the definition of $\phi_\epsilon$ we have

$$\Pr_{S \leftarrow \mu^m} [|\mu(\mathcal{E}_\epsilon) - \hat{\mu}_S(\mathcal{E}_\epsilon)| \geq \delta] \leq \phi(m, \delta) + \phi_\epsilon(m, \delta). \tag{8}$$

Combining Equations 7 and 8, by union bound we get

$$\Pr_{S \leftarrow \mu^m} [\mu(\mathcal{E}_\epsilon) \leq h(\hat{\mu}_S, \alpha, \epsilon, \mathcal{G}) - \delta] \leq \phi(m, \delta) + \phi_\epsilon(m, \delta)$$

which by the definition of $\mathcal{E}$ implies

$$\Pr_{S \leftarrow \mu^m} [h(\mu, \alpha + \delta, \epsilon, \mathcal{G}) \leq h(\hat{\mu}_S, \alpha, \epsilon, \mathcal{G}) - \delta] \leq \phi(m, \delta) + \phi_\epsilon(m, \delta). \tag{9}$$

Now combining Equations 6 and 9, by union bound we have

$$\Pr_{S \leftarrow \mu^m} [h(\mu, \alpha - \delta, \epsilon, \mathcal{G}) - \delta \leq h(\hat{\mu}_S, \alpha, \epsilon, \mathcal{G}) \leq h(\mu, \alpha + \delta, \epsilon, \mathcal{G}) + \delta] \geq 1 - 2\left(\phi(m, \delta) + \phi_\epsilon(m, \delta)\right).$$

$\square$

## A.2 Proof of Theorem 3.5

In this section, we prove Theorem 3.5 using ideas similar to ideas used in Scott & Nowak (2006). Before proving the theorem, we lay out the following lemma which will be used in the proof.

**Lemma A.1** (Borel-Cantelli Lemma). *Let $\{E_T\}_{T \in \mathbb{N}}$ be a series of events such that*

$$\sum_{T=1}^{\infty} \Pr[E_T] < \infty$$

*Then with probability 1, only finite number of events will occur.*

Now we are ready to prove Theorem 3.5.

*Proof of Theorem 3.5.* Define $E_T$ to be the event that
$h(\mu, \alpha - \delta(T), \epsilon, \mathcal{G}(T)) - \delta(T) > h(\hat{\mu}_{S_T}, \alpha, \epsilon)$ or $h(\mu, \alpha + \delta(T), \epsilon, \mathcal{G}(T)) + \delta(T) < h(\hat{\mu}_{S_T}, \alpha, \epsilon, \mathcal{G})$.
Based on Theorem 3.3 we have $\Pr[E_T] \leq 2 \cdot (\phi^T(m(T), \delta(T)) + \phi_{\epsilon}^T(m(T), \delta(T)))$. Therefore, by Conditions 1 and 2 we have

$$\sum_{T=1}^{\infty} \Pr[E_T] \leq 2 \left( \sum_{T=1}^{\infty} \phi^T(m(T), \delta(T)) + \phi_{\epsilon}^T(m(T), \delta(T)) \right) < \infty.$$

Now by Lemma A.1, we know there exist with measure 1 some $j \in \mathbb{N}$, such that for all $T \geq j$,

$$h(\mu, \alpha - \delta(T), \epsilon, \mathcal{G}(T)) - \delta(T) \leq h(\hat{\mu}_{S_T}, \alpha, \epsilon, \mathcal{G}(T)) \leq h(\mu, \alpha + \delta(T), \epsilon, \mathcal{G}(T)) + \delta(T).$$

The above implies that

$$\lim_{T \to \infty} h(\mu, \alpha - \delta(T), \epsilon, \mathcal{G}(T)) - \delta(T) \leq \lim_{T \to \infty} h(\hat{\mu}_{S_T}, \alpha, \epsilon, \mathcal{G}(T)) \leq \lim_{T \to \infty} h(\mu, \alpha + \delta(T), \epsilon, \mathcal{G}(T)) + \delta(T).$$

We know that

$$\lim_{T \to \infty} h(\mu, \alpha - \delta(T), \epsilon, \mathcal{G}(T)) = \lim_{T_1 \to \infty} \lim_{T_2 \to \infty} h(\mu, \alpha - \delta(T_1), \epsilon, \mathcal{G}(T_2))$$

$$(\text{By condition 4}) \quad = \lim_{T_1 \to \infty} h(\mu, \alpha - \delta(T_1), \epsilon)$$

$$(\text{By local continuity and condition 3}) \quad = h(\mu, \alpha, \epsilon).$$

Similarly, we have

$$\lim_{T \to \infty} h(\mu, \alpha + \delta(T), \epsilon, \mathcal{G}(T)) = h(\mu, \alpha, \epsilon).$$

Therefore we have,

$$\lim_{T \to \infty} h(\mu, \alpha, \epsilon) - \delta(T) \leq \lim_{T \to \infty} h(\hat{\mu}_{S_T}, \alpha, \epsilon, \mathcal{G}(T)) \leq \lim_{T \to \infty} h(\mu, \alpha, \epsilon) + \delta(T)$$

which by condition 3 implies

$$\lim_{T \to \infty} h(\hat{\mu}_{S_T}, \alpha, \epsilon, \mathcal{G}(T)) = h(\mu, \alpha, \epsilon).$$

$\square$

## A.3 Proof of Theorem 3.8

*Proof.* This theorem follows from our general Theorem 3.5. We show that the choice of parameters here satisfies all four conditions of Theorem 3.5.

If we let $\mathcal{G}(T)$ to be the collection of subsets specified by complement of union of $T$ hyperrectangles. Then $\mathcal{G}_{\epsilon}(T)$ will be the collection of of subsets specified by complement of union of $T$ hyperrectangles that are bigger than $\epsilon$ in each coordinate. Therefore we have $\mathcal{G}_{\epsilon}(T) \subset \mathcal{G}(T)$. We know that the VC dimension of $\mathcal{G}(T)$ is $d_T = O(nT \log(T))$ because the VC dimension of all hyperrectangles is $O(n)$ and the functions formed by $T$ fold union of functions in a VC class is at most $n \cdot T \log(T)$ (See Eisenstat & Angluin (2007)). Therefore, by VC inequality we have

$$\Pr_{S \leftarrow \mu^m} \left[ \sup_{\mathcal{E} \in \mathcal{G}(T)} |\mu(\mathcal{E}) - \hat{\mu}_S(\mathcal{E})| \geq \delta \right] \leq 8 e^{nT \log(T) \log(m) - m\delta^2/128}.$$

Therefore $\Phi^T(m, \delta) = 8 e^{nT \log(T) \log(m) - m\delta^2/128}$ is a complexity penalty for both $\mathcal{G}(T)$ and $\mathcal{G}_{\epsilon}(T)$. Hence, if we define $\delta(T) = 1/T$ and $m(T) \geq T^4$, then the first three conditions of Theorem 3.5 are satisfied. The fourth condition is also satisfied by the universal consistency of histogram rules (See Devroye et al. (2013), Ch. 9). $\square$

### A.4 Proof of Theorem 3.10

*Proof.* Similar to Theorem 3.8 This theorem follows from our general Theorem 3.5. We show that the choice of parameters here satisfies all four conditions of Theorem 3.5.

If we let $\mathcal{G}(T)$ to be the collection of subsets specified by union of $T$ balls. Then $\mathcal{G}_\epsilon(T)$ will be the collection of of subsets specified by union of $T$ balls with diameter at least $\epsilon$. Similar to the proof of Theorem 3.8, we have $\mathcal{G}_\epsilon(T) \subset \mathcal{G}(T)$. We know that the VC dimension of all balls is $O(n)$ so using the fact that $\mathcal{G}(T)$ is $T$ fold union of balls, the VC dimension of $\mathcal{G}(T)$ is $d_T = O(nT \log(T))$ (See Eisenstat & Angluin (2007)). Therefore, by VC inequality we have complexity penalties similar to those of Theorem 3.8 for both $\mathcal{G}(T)$ and $\mathcal{G}_\epsilon(T)$. Hence, if we define $\delta(T) = 1/T$ and $m(T) \geq T^4$, then the first three conditions of Theorem 3.5 are satisfied. The fourth condition is also satisfied by the universal consistency of kernel-based rules (See Devroye et al. (2013) , Ch. 10).  $\square$

## B   The Proposed Algorithms

This section provides the pseudocode and a runtime analysis for our algorithms for finding robust error regions under $\ell_\infty$ and $\ell_2$, respectively.

### B.1   Pseudocode

---
**Algorithm 1:** Heuristic Search for Robust Error Region under $\ell_\infty$

---
**Input**  : a set of images $\mathcal{S}$; perturbation strength $\epsilon_\infty$; error threshold $\alpha$; number of hyperrectangles $T$; number of nearest neighbours $k$; precision for binary search $\delta_{\text{bin}}$.

1  $r_k(\boldsymbol{x}) \leftarrow$ compute the $\ell_1$-norm distance to the $k$-th nearest neighbour for each $\boldsymbol{x} \in \mathcal{S}$;
2  $\mathcal{S}_{\text{sort}} \leftarrow$ sort all the images in $\mathcal{S}$ by $r_k(\boldsymbol{x})$ in an ascending order;
3  $q_{\text{lower}} \leftarrow 0.0$,  $q_{\text{upper}} \leftarrow 1.0$;
4  **while** $q_{upper} - q_{lower} > \delta_{bin}$ **do**
5  $\quad$ $q \leftarrow (q_{\text{lower}} + q_{\text{upper}})/2$;
6  $\quad$ perform kmeans clustering algorithm ($T$ clusters, $\ell_1$ metric) on the top-$q$ images of $\mathcal{S}_{\text{sort}}$;
7  $\quad$ $\{\boldsymbol{u}^{(t)}\}_{t=1}^T \leftarrow$ record the centroids of the resulted $T$ clusters;
8  $\quad$ **for** $t = 1, 2, \ldots, T$ **do**
9  $\quad\quad$ $\mathcal{R}ect(\boldsymbol{u}^{(t)}, \boldsymbol{r}^{(t)}) \leftarrow$ cover $t$-th cluster with the minimum-sized rectangle centered at $\boldsymbol{u}^{(t)}$;
10 $\quad$ **end**
11 $\quad$ $\mathcal{E}_q \leftarrow \mathcal{X} \setminus \cup_{t=1}^T \mathcal{R}ect_{\epsilon_\infty}(\boldsymbol{u}^{(t)}, \boldsymbol{r}^{(t)})$;  // $\mathcal{R}ect_\epsilon(\boldsymbol{u}, \boldsymbol{r})$  denotes the $\epsilon$-expansion of $\mathcal{R}ect(\boldsymbol{u}, \boldsymbol{r})$
12 $\quad$ **if** $|\mathcal{S} \cap \mathcal{E}_q|/|\mathcal{S}| \geq \alpha$ **then**
13 $\quad\quad$ $q_{\text{lower}} \leftarrow q$,  $AdvRisk_q \leftarrow \big|\{\boldsymbol{x} \in \mathcal{S} : \boldsymbol{x} \notin \cup_{t=1}^T \mathcal{R}ect(\boldsymbol{u}^{(t)}, \boldsymbol{r}^{(t)})\}\big|/|\mathcal{S}|$;
14 $\quad$ **else**
15 $\quad\quad$ $q_{\text{upper}} \leftarrow q$;
16 $\quad$ **end**
17 **end**
18 $\hat{q} \leftarrow \arg\min_q \{AdvRisk_q\}$;
   **Output :** $(\hat{q}, AdvRisk_{\hat{q}}, \mathcal{E}_{\hat{q}})$

---

### B.2   Runtime Analysis

For $\ell_\infty$, we construct the systems of hyperrectangles by first precomputing an approximate k-NN distance estimate using Ball Trees (Omohundro, 1989; Pedregosa et al., 2011) for each data point, and then clustering the top-$q$ densest data points into $T$ partitions using the k-means algorithm, where we binary search for the optimal parameter $q$. The time complexity of precomputing and sorting the nearest neighbor distance estimates is approximately $O(nd \log(n))$, where $n$ is the total number of data points in $\mathbb{R}^d$. In addition, the time complexity of k-means algorithm is $O(ndTI)$, where $I$ is the averaged number of iterations for k-means algorithm to converge. Therefore, the total time complexity of the proposed algorithm for $\ell_\infty$ is $O(nd \log(n) + ndTI \log(1/\delta))$. In our experiments on CIFAR-10 ($\epsilon_\infty = 8/255$, $T = 40$ and $\delta = 0.005$), the proposed algorithm takes 76 minutes for

**Algorithm 2:** Heuristic Search for Robust Error Region under $\ell_2$

---

**Input** : a set of images $\mathcal{S}$; perturbation strength $\epsilon_2$; error threshold $\alpha$; number of balls $T$.

**1** $\hat{\mathcal{E}} \leftarrow \{\}, \quad \hat{\mathcal{S}}_{\text{init}} \leftarrow \{\}, \quad \hat{\mathcal{S}}_{\text{exp}} \leftarrow \{\}$;

**2** **for** $t = 1, 2, \ldots, T$ **do**

**3** $\quad k_{\text{lower}} \leftarrow \lceil (\alpha|\mathcal{S}| - |\hat{\mathcal{S}}_{\text{init}}|)/(T - t + 1) \rceil, \quad k_{\text{upper}} \leftarrow (\alpha|\mathcal{S}| - |\hat{\mathcal{S}}_{\text{init}}|)$;

**4** $\quad$ **for** $u \in \mathcal{S}$ **do**

**5** $\quad\quad$ **for** $k \in [k_{lower}, k_{upper}]$ **do**

**6** $\quad\quad\quad r_k(u) \leftarrow$ compute the $\ell_2$ distance from $u$ to the $k$-th nearest neighbour in $\mathcal{S} \setminus \hat{\mathcal{S}}_{\text{init}}$;

**7** $\quad\quad\quad \mathcal{S}_{\text{init}}(u, k) \leftarrow \{x \in \mathcal{S} \setminus \hat{\mathcal{S}}_{\text{init}} : \|x - u\|_2 \leq r_k(u)\}$;

**8** $\quad\quad\quad \mathcal{S}_{\text{exp}}(u, k) \leftarrow \{x \in \mathcal{S} \setminus \hat{\mathcal{S}}_{\text{exp}} : \|x - u\|_2 \leq r_k(u) + \epsilon_2\}$;

**9** $\quad\quad$ **end**

**10** $\quad$ **end**

**11** $\quad (\hat{u}, \hat{k}) \leftarrow \text{argmin}_{(u,k)}\{|\mathcal{S}_{\text{exp}}(u, k)| - |\mathcal{S}_{\text{init}}(u, k)|\}$;

**12** $\quad \hat{\mathcal{E}} \leftarrow \hat{\mathcal{E}} \cup \text{Ball}(\hat{u}, r_{\hat{k}}(\hat{u}))$;

**13** $\quad \hat{\mathcal{S}}_{\text{init}} \leftarrow \hat{\mathcal{S}}_{\text{init}} \cup \mathcal{S}_{\text{init}}(\hat{u}, \hat{k}), \quad \hat{\mathcal{S}}_{\text{exp}} \leftarrow \hat{\mathcal{S}}_{\text{exp}} \cup \mathcal{S}_{\text{exp}}(\hat{u}, \hat{k})$;

**14** **end**

**Output** : $\hat{\mathcal{E}}$

---

precomputing the nearest neighbors, and takes around 2 hours for the iterative steps to converge on a Intel Xeon CPU E5-2620 v4 server with 32 processors.

For $\ell_2$, instead of computing the k-NN distances for each iteration, we precompute and keep the k-NN neighbours using Ball Trees for each image to save computation, which requires a time complexity of $O(nd \log n)$. The iterative steps require the major computation of $O(\alpha T n^2 d)$, since we iterate through all the possible choices of ball centers and corresponding radii to find the optimal error region with the smallest expansion. We believe the quadratic dependency on the sample size can be improved using better searching algorithm for finding the robust error region. Since our main focus is to understand the limitation of robust learning on real datasets, we leave the optimization of the proposed heuristic method for better computational efficiency as future work.

## C   Other Experimental Results

### C.1   Results for $\ell_\infty$ on other datasets

We also evaluate the proposed empirical method for $\ell_\infty$ metric on other benchmark image datasets, including Fashion-MNIST (Xiao et al., 2017) and SVHN (Netzer et al., 2011).

Table 4: Summary of the main results using our method for different settings with $\ell_\infty$ perturbations.

| Dataset | $\alpha$ | $\epsilon_\infty$ | $T$ | Best $q$ | Empirical Risk (%) | | Empirical AdvRisk (%) | |
|---|---|---|---|---|---|---|---|---|
| | | | | | training | testing | training | testing |
| Fashion-MNIST | 0.05 | 0.1 | 10 | 0.758 | $5.64 \pm 0.78$ | $5.92 \pm 0.85$ | $10.30 \pm 0.72$ | $11.56 \pm 0.84$ |
| | | 0.2 | 10 | 0.726 | $5.79 \pm 1.00$ | $6.00 \pm 1.02$ | $13.44 \pm 0.60$ | $14.82 \pm 0.71$ |
| | | 0.3 | 10 | 0.668 | $5.90 \pm 0.94$ | $6.13 \pm 0.93$ | $17.46 \pm 0.53$ | $18.87 \pm 0.66$ |
| SVHN | 0.05 | 0.01 | 10 | 0.812 | $5.21 \pm 0.19$ | $8.83 \pm 0.30$ | $6.08 \pm 0.20$ | $10.17 \pm 0.29$ |
| | | 0.02 | 10 | 0.773 | $5.31 \pm 0.12$ | $8.86 \pm 0.20$ | $7.76 \pm 0.12$ | $12.46 \pm 0.15$ |
| | | 0.03 | 10 | 0.750 | $5.15 \pm 0.13$ | $8.55 \pm 0.22$ | $8.88 \pm 0.13$ | $13.82 \pm 0.25$ |

### C.2   Detailed results for $\ell_2$ using our method

In this section, we demonstrate the detailed training and testing results on the best error region obtained using Algorithm 2 on MNIST and CIFAR-10 with $\ell_2$ perturbations, as well as results on Fashion-MNIST and SVHN. Note that for the additional datasets, we set $\alpha$ to be the same as the case of $\ell_\infty$ and set $\epsilon_2 = \sqrt{n/\pi} \cdot \epsilon_\infty$ using the same conversion rule, where $n$ is the input dimension.

Table 5: Summary of the main results using our method for different settings with $\ell_2$ perturbations.

| Dataset | $\alpha$ | $\epsilon_2$ | $T$ | Empirical Risk | | Empirical AdvRisk | |
|---|---|---|---|---|---|---|---|
| | | | | training | testing | training | testing |
| MNIST | 0.01 | 1.58 | 20 | 1.25% | 1.07% | 2.23% | 2.19% |
| | | 3.16 | 20 | 1.25% | 1.02% | 4.35% | 4.15% |
| | | 4.74 | 20 | 1.25% | 1.07% | 10.71% | 10.09% |
| CIFAR-10 | 0.05 | 0.2453 | 5 | 5.00% | 5.16% | 5.22% | 5.53% |
| | | 0.4905 | 5 | 5.00% | 5.14% | 5.61% | 5.83% |
| | | 0.9810 | 5 | 5.00% | 5.12% | 6.38% | 6.56% |
| Fashion-MNIST | 0.05 | 1.58 | 10 | 5.25% | 5.07% | 7.84% | 7.77% |
| | | 3.16 | 10 | 5.25% | 4.99% | 15.95% | 16.23% |
| | | 4.74 | 10 | 5.25% | 5.21% | 19.76% | 20.10% |
| SVHN | 0.05 | 0.3127 | 10 | 5.00% | 6.92% | 5.24% | 7.34% |
| | | 0.6254 | 10 | 5.00% | 7.30% | 5.59% | 8.16% |
| | | 0.9381 | 10 | 5.00% | 7.56% | 5.96% | 8.94% |

## C.3 Additional training curves

(a) MNIST ($\epsilon_\infty = 0.1$ and $T = 10$)

(b) MNIST ($\epsilon_\infty = 0.3$ and $T = 10$)

(c) CIFAR-10 ($\epsilon_\infty = 4/255$ and $T = 20$)

(d) CIFAR-10 ($\epsilon_\infty = 8/255$ and $T = 40$)

Figure 2: Risk and adversarial risk of the corresponding region as $q$ varies under different settings.

(a)

(b)

Figure 3: Adversarial risk of the resulted error region with best $q$ obtained using our method as $T$ varies under different settings: (a) MNIST ($\epsilon = 0.1$, $\alpha = 0.01$) and CIFAR-10 ($\epsilon_\infty = 2/255$, $\alpha = 0.05$); (b) MNIST ($\epsilon_\infty = 0.2$, $\alpha = 0.01$) and CIFAR-10 ($\epsilon_\infty = 4/255$, $\alpha = 0.05$)