[Reviews · NeurIPS 2019]

Reviewer 1



I doubt the applicability of the work to real-world problems. It involves constructing systems of rectangles around points in input space, a potentially tedious task. The authors should explain more how this is actually done, and why it doesn't represent a computational bottleneck for the proposal. I didn't check the technical details in the paper, but they appear sound from afar. Typos: 165: Typo "been been" --> "been" line 174: series ==> sequences

Reviewer 2



This work asks the following fundamental question about ML classifiers and data distributions: how close is the nearest in-distribution error? Test error, as measured by a random iid sample from the data distribution provides an estimate of the volume of errors in that distribution. Given that this volume exists there must be a nearest error to a given input within that volume. Counter-intuitively, for many high dimensional data distributions the nearest error is expected to be significantly closer than the nearest sampled error in the test set. For several synthetic distributions, rigorous bounds can be proved which imply that small adversarial perturbations must exist for any model with a non-vanishing error rate on the test set---a phenomenon known as concentration of measure. A fundamental open question since the publication of https://arxiv.org/abs/1801.02774, is whether or not similar bounds can be expected to hold on real-world data distributions. This work takes the first steps towards proving such bounds. Proving such a bound is extremely difficult for traditional ML datasets because the underlying p(x) is unknown, one only has access to a finite sample. Remarkably, the authors have managed to develop a general theory which works with just a finite sample and any metric on the input space. Summarizing the proof technique is difficult as it is somewhat involved and requires many careful definitions. To be honest, in the limited time I had to review this paper I was not able to check the proofs in detail. However, the definitions are clear, and the overall proof strategy is natural yet very clever. While the result is general, it requires constructing a sequence of subset collections on the input space which match certain properties. Verifying that these properties hold is computationally prohibitive, and so the authors provide an approximate algorithm, which means the resulting bounds may also be approximate. The authors use their theory to prove bounds on adversarial robustness on the MNIST and CIFAR-10 datasets. Perhaps the strongest bound they prove is that a 6% error rate on CIFAR-10 implies that l infinity 16/255 robustness cannot exceed 71%. This result is quite remarkable, and I think the authors have undersold the significance. The most recent claimed SOTA on CIFAR-10 is 57% robustness (8/255) at 13% natural error rate (https://arxiv.org/pdf/1901.09960.pdf). While there still is a gap, this is suggestive that recent work may be approaching fundamental limits on adversarial robustness in terms of imperfect generalization. It is important to note that the authors make no assumption about the shape of the error set, it seems likely that by making additional assumptions about the error set (given that it must arise from an ML classifier) that one could close the gap even further. Additionally, l_p perturbations are only a toy threat model with little to no real world significance. Actual adversaries rarely restrict themselves to tiny perturbations of an input, see for example this documented attack on YouTube's ContentID system https://qz.com/721615/smart-pirates-are-fooling-youtubes-copyright-bots-by-hiding-movies-in-360-degree-videos/. With all of this in mind, this work does raise questions as to why the research community has invested so much time and effort specifically on the problem of l_p perturbations when there is still much work to be done improving model generalization (and in particular o.o.d robustness https://arxiv.org/abs/1901.10513). Specific comments/questions: It is worth including a discussion on https://arxiv.org/abs/1805.12152, which many researchers point to as refuting the concentration of measure hypothesis. Empirically, it is the case that adversarial training does degrade generalization while improving l_p robustness. However, this is not at odds with concentration of measure. While there is a sizeable gap between naturally trained models and any fundamental bound on robustness based on concentration of measure, this theory may explain why adversarially trained models still lack robustness at larger epsilon. I would also be a bit more specific in the introduction that lp perturbations are a toy threat model that is not intended to model how actual adversaries choose to break systems. The popularity of lp-robustness has, in my opinion, distracted the research community from focusing on the more pressing issue: distribution shift (https://arxiv.org/abs/1807.01697, https://arxiv.org/abs/1907.07174). Statement of Theorem 3.3 (and other places): The notation for product measure was not defined. Definition 2.1: Any reason the authors defined an adversarial example as the nearest input which is classified differently, and not the nearest error? While subtle, the second definition is more general and will become necessary as robustness improves. For example, on the concentric spheres distribution it was observed that in some settings the nearest error was actually farther than the nearest input of the other class. Many adversarial defense papers declare success when the optimization algorithm starts producing inputs that visually look like the other class, but it is usually the case that if one searches differently, adversarial perturbations can still be found. See for example the background attack in Figure 6 of https://arxiv.org/pdf/1807.06732.pdf. Table 1: It's worth noting that MNIST may be a degenerate case with respect to the l_infinity metric. In particular, a trivial defense is to first threshold the inputs about .5 and classify the resulting binary image. Because of this, I would not expect any meaningful bounds to hold for this dataset and metric.

Reviewer 3



Several recent theoretical papers aiming to explain the "adversarial examples" phenomenon have suggested that adversarial examples are an inevitable consequence of (1) concentration of measure, and (2) non-zero generalization error. In more detail: "concentrated" probability distributions have the property that the epsilon-expansion of *any* set -- even one with small probability measure -- is guaranteed to have very large probability measure. Now, for any classifier, define the error region to be the set of points where the classifier is wrong. The standard generalization error of a classifier is the measure of the error region under the data distribution, and the robust generalization error is the measure of the epsilon-expansion of the error region. Therefore, if the data distribution is concentrated, then small but nonzero standard error rates would imply large robust error rates. However, it has remained unclear whether actual distributions of interest, such as the CIFAR-10 distribution over natural images, are sufficiently concentrated to the extent that concentration of measure could fully explain the adversarial examples phenomenon. One reason is that there are no known tools for estimating the concentration function of a distribution given just an i.i.d sample from that distribution. This paper claims two main contributions, one theoretical and one empirical: (1) a method for estimating a distribution's concentration function given just an i.i.d sample (2) the experimental finding that MNIST and CIFAR-10 are *not* sufficiently concentrated so as to explain adversarial vulnerability. Summary of contribution #1: the two main challenges in estimating the concentration function are (1) we only have access to the empirical data distribution, not the true data distribution, and (2) even if we knew the true data distribution, estimating the concentration would require performing a computationally intractable optimization over all sets. The authors propose to get around these challenges by instead estimating the concentration function restricted to a statistically "simple" family of sets -- for example, the sets that are the complement of a union of T hyper-rectangles, or the sets that are a union of T L2 balls. This solves both problems. First, optimizing over a "simple" family of sets is (sort of...) more computationally feasible than optimizing over _all_ sets. Second, if this family of sets is sufficiently "simple" -- more specifically, if it obeys a uniform convergence law -- then the concentration of the true data distribution (which we don't have access to) is guaranteed w.hp. to be close to the concentration of the empirical data distribution (which we do). That is the first theoretical result (Theorem 3.3). However, this gives us the concentration function over the "simple" family of sets, not the actual concentration function (though the former is by definition an upper bound on the latter). This paper's second theoretical result (Theorem 3.5) is that as the number of samples goes to infinity, if we increase the complexity of this "simple" family in a slow enough way, then the concentration function over the "simple" family of sets converges to the actual concentration function. Comments on contribution #1: this is a theoretical contribution which does not seem very useful in practice. First, it's not *actually* computationally tractable to optimize exactly over e.g. "the family of all sets that are a union of T L2 balls," so the authors have to resort to heuristics. Second, the restricted concentration function only converges to the true concentration function in the asymptotic limit of infinite data, and there are no bounds which tell us how close we are. Thus, the theoretical innovations in this paper are not practically relevant to the study of adversarial vulnerability. I am not personally qualified to assess the standalone contribution of these theoretical results to the field of statistics. More generally, the "adversarial examples are inevitable" theory posits a *lower bound* on the concentration function, i.e. a finding that every set with measure alpha has an epsilon-expansion with measure at least X. Therefore, to disprove the "adversarial examples are inevitable" theory, you only need to show an *upper bound* on the concentration function, i.e. a finding that there exists some set with measure alpha whose epsilon-expansion has measure at most Y. Given a sample from the data distribution, here is a simple way to do that: split the sample into a "training set" and a "test set" (loosely speaking), use the training set + any heuristic search algorithm to search for some set with measure alpha under the training set whose epsilon-expansion has low measure under the training set, and use the measure under the *test set* of this set's epsilon expansion, in conjunction with standard concentration bounds, to show an upper bound on the measure of this set's epsilon expansion under the true data distribution. Basically, since we can split the sample into a training partition (which we use to search for the set) and a test partition (which we use to estimate the measure of the set's epsilon expansion), there is no need to invoke uniform convergence. Now, this is exactly what the authors do, in the second part of the paper. But I think it could be made clearer that the correctness of this procedure does not depend *in any way* on the theory in the first part of the paper. (From the authors' perspective, this is both good, since it means that their paper's experimental contribution does not rely on any of the sketchy asymptotic theory, and bad, since it renders somewhat useless the theory in the first part.) Summary of contribution #2: the authors propose heuristic algorithms for finding sets whose measure under the empirical data distribution is alpha, but whose epsilon-expansion (taken under either the L-inf or the L2 norm) is as small as possible. In the L-inf case, these sets are the complement of a union of hyperrectangles. In the L2 case, the sets are the union of balls. Using this method, the authors find that for CIFAR-10 there exists a set with measure 5.94% whose epsilon-expansion under L-inf norm with radius 8/255 has measure 18.13%. In contrast, the robust error at that radius of the SOTA certified L-inf classifier (from Wong et al, "Scaling Provable Adversarial Defenses") is 70%. Likewise, there exists a set with measure 6.36% whose epsilon-expansion under L2 norm with radius 0.98 has measure 6.56%. In contrast, the robust error at that radius of the SOTA certified L2 classifier (from Salman et al, "Provably Robust Deep Learning via Adversarially Trained Smoothed Classifiers") is 64%. This means that concentration of measure cannot fully explain the high error rates we currently see for robust classifiers. As a sidenote, while the authors interpret this to mean that there is room to develop better robust classifiers, it could also mean that robustness is impossible for reasons other than concentration of measure. Comments on contribution #2: this is a useful contribution to the literature. First, I like the heuristic algorithms for optimizing over these restricted families of sets. The nice thing about these restricted set families (apart from the fact, emphasized in the paper, that they have finite VC dimension) is that you can *exactly* compute the (empirical) measure of their epsilon-expansions, without resorting to non-convex optimization like PGD adversarial attacks. Second, this paper provides convincing evidence against the "adversarial examples are caused by concentration of measure" hypothesis which has surfaced lately in the literature. ===== originality: the work is highly original quality: the work is of high quality clarity: the authors should be more explicit about the fact (which I mentioned above) that their experiments do not rely in any way on the theory part of their paper. significance: the experimental results are significant to the study of adversarial robustness, as they disprove one of the hypotheses (concentration of measure) which purported to explain adversarial vulnerability. In addition, the heuristic algorithms -- for finding sets of fixed measure whose epsilon-expansions in various norms have small measure -- are a useful independent contribution. The theoretical results in this paper are not significant to the study of adversarial robustness, though they may be significant in general to the field of statistics (but I'm not qualified to assess that). === Update: in their response, the authors confirmed that the paper's theory is not necessary to prove the *correctness* of the results in Tables 1&2, or of the more general finding that concentration of measure is an insufficient explanation for adversarial vulnerability. However, they pointed out that the theory was helpful in deriving the algorithm that enabled them to come up with the results in Tables 1&2. (That is: if they had searched over a more complex family of sets, then the gap between train and test empirical measure would have probably been larger.) I encourage the authors to make both of these points explicit in the revision. I suspect that, as is, many readers will see the asymptotic conditions in Theorem 3.5 and assume that the results in Tables 1&2 (and, more generally, the paper's main conclusion) depend on these assumptions.

[Author Response · NeurIPS 2019]

We thank all the reviewers for their thorough reviews and insightful comments. Reviewers' comments are in blue.

**Reviewer 1:** *The authors should explain more how this is actually done, and why it doesn't represent a computational bottleneck for the proposal.* The detailed algorithms for finding robust error regions under $\ell_\infty$ and $\ell_2$ perturbations are provided in Section C in our supplementary materials. For $\ell_\infty$, we construct the systems of hyperrectangles by first precomputing an approximate k-NN distance estimate using Ball Trees for each data point, and then clustering the top-$q$ densest data points into $T$ partitions using the k-means algorithm, where we binary search for the optimal parameter $q$. The time complexity of precomputing and sorting the nearest neighbor distance estimates is approximately $O(nd\log(n))$, where $n$ is the total number of data points in $\mathbb{R}^d$. In addition, the time complexity of k-means algorithm is $O(ndTI)$, where $I$ is the averaged number of iterations for k-means algorithm to converge. Therefore, the total computational complexity of our algorithm for $\ell_\infty$ is $O(nd\log(n) + ndTI\log(1/\delta))$, where $\delta$ is the stopping threshold for binary search. In our experiments, we applied our algorithms to medium-sized datasets including CIFAR-10 and SVHN, and they finished reasonably quickly. We will include a runtime analysis of algorithms in the final version.

**Reviewer 2:** *It is worth including a discussion on https://arxiv.org/abs/1805.12152, which many researchers point to as refuting the concentration of measure hypothesis.* We will include the discussions. In a nutshell, that work uses a definition that coincides with the definition of adversarial examples that we use for the interesting range of tampering parameters (in which the ground truth is robust). But, if the tampering goes up and can change the ground truth, even learning a concept *exactly* might leave room for adversarial examples under the definition used in that work (but not under ours). *I would also be a bit more specific in the introduction that $l_p$ perturbations are a toy threat model that is not intended to model how actual adversaries choose to break systems.* We will make sure to add comments about shortcomings of $l_p$ norm in capturing the whole picture. *Any reason the authors defined an adversarial example as the nearest input which is classified differently, and not the nearest error?* We agree with you that nearest error point is a natural definition of adversarial examples and we indeed use this definition in the paper. Specifically, in Definition 2.1, we compare the true label of $x'$ with the predicted label of $x'$, which means $x'$ should be an error point to be counted as an adversarial example. We will make this point more clear in the statement of our definition. *It's worth noting that MNIST may be a degenerate case with respect to the $l_\infty$ metric. In particular, a trivial defense is to first threshold the inputs about .5 and classify the resulting binary image. Because of this, I would not expect any meaningful bounds to hold for this dataset and metric.* We agree with the reviewer that thresholding MNIST (and any other dataset) will make the transformed distribution not concentrated under $l_\infty$. However, the original distribution (before transformation) might still be concentrated. In particular, one might be able to add a small perturbation to the image before thresholding the features and make the binary transformation of the perturbed image different from that of the original image. For the case of MNIST, it seems that binarizing images should not change the distribution much, as the original images have close to binary form. Our experiments support this intuition and show that regions in MNIST dataset could have a very small expansion (it only grows from $\sim 1\%$ to $\sim 10\%$ when allowing $\epsilon = 0.4$ perturbations). *It would be very interesting if the authors could strengthen their bounds by making additional assumptions on the shape of the error set. Additionally, one could strengthen the bounds by approximating the content-preserving threat model.* Thank you for pointing out these interesting future directions, particularly for the content-preserving mode. Interestingly, part of our theoretical results do already prove such results for restricted forms of error sets, and this does set the stage on how we choose the sets for our experiments. The *proof* of Theorem 3.5 first proves such result for limited shapes. In particular, we obtain such result when VC dimension of the sets *and* their expansion are bounded (e.g., union of hyperrectangles).

**Reviewer 3:** *Q1- the theoretical innovations in this paper are not practically relevant to the study of adversarial vulnerability. Q2- to disprove the "adversarial examples are inevitable" theory, you only need to show an \*upper bound\* on the concentration function, i.e. a finding that there exists some set with measure alpha whose epsilon-expansion has measure at most Y. Given a sample from the data distribution, here is a simple way to do that: split the sample into a "training set" and a "test set".* The two questions/comments are relevant. We start with Q2, then will address Q1 as well. Yes, indeed to show that a distribution does not concentrate beyond a parameter, one can aim to show the existence of *some* set (found based on "training set") and test its expansion using the "test set". However, the question is how to design algorithms that come up with such sets. Our theory tells us that by looking at specific types of sets (e.g., collection of hyperrectangles), we can get "generalization" bounds for estimating expansion. Note that we tried different collections of subsets (e.g., subsets decided by neural networks) that were not supported by our theory and we observed huge generalization error that made the experiment meaningless. Therefore, in our experiments we use exactly the subset collections that theory suggests and the results of experiments verify our theory. Our theory is also important for *future* work. If one wants to find the concentration of measure under another metric probability space, they can use our theory to come up with suitable subset collections with generalization guarantees. *As a sidenote, while the authors interpret this to mean that there is room to develop better robust classifiers, it could also mean that robustness is impossible for reasons other than concentration of measure.* Thank you for pointing this out. We tried to be cautious in interpreting our results and consider the "robust classification is impossible for other reasons" hypothesis. However, after reading through the paper we found an occasion in our discussions (line 284-285) that we did not consider this hypothesis. We will make sure to clarify this in the next version of our paper.

[Meta-Review · NeurIPS 2019]

All reviewers agreed that this is a originaly and solid contribution to NeurIPS.